# Using Nano Zero-Valent Iron Supported on Diatomite to Remove Acid Blue Dye: Synthesis, Characterization, and Toxicology Test

Ernesto Flores-Rojas [1], Denhi Schnabel [2], Erick Justo-Cabrera [2], Omar Solorza-Feria [3] , Héctor M. Poggi-Varaldo [4] and Luz Breton-Deval [2,5,*]

1    Campus San Rafael, Universidad del Valle de México, Mexico City 06740, Mexico; erefrojas@gmail.com
2    Instituto de Biotecnología, Universidad Nacional Autónoma de México, Cuernavaca 62210, Mexico; denhi.schnabel@ibt.unam.mx (D.S.); erickjustoc@gmail.com (E.J.-C.)
3    Programa de Nanociencias y Nanotecnología, Centro de Investigación y de Estudios Avanzados del Instituto Politécnico Nacional, Mexico City 07360, Mexico; osolorza@cinvestav.mx
4    Environmental Biotechnology and Renewable Energies Group, Department of Biotechnology and Bioengineering, CINVESTAV-IPN, Mexico City 07360, Mexico; lazarillodetormes1001@gmail.com
5    Consejo Nacional de Ciencia y Tecnología, Mexico City 03940, Mexico
*    Correspondence: luz.breton@ibt.unam.mx

**Abstract:** This work aimed to synthesize and characterize nanoscale zero-valent iron (nZVI), supported on diatomaceous earth (DE) at two different molar concentrations, 3 and 4 M (nZVI-DE-1 nZVI-DE-2), to test the decolorization treatment of acid blue dye (AB) and perform a toxicological test using zebrafish. The synthesis of the nanoparticles was obtained using the chemical reduction method. The material was fully characterized by X-ray diffraction, scanning electron microscopy (SEM), energy-dispersive X-ray (EDX), and transmission electron microscopy and specific surface area (BET). The results showed spherical forms in clusters between 20 and 40 nm of zero-valent iron supported on diatomaceous earth. The removal of 1 g/L of AB from water treated with nZVI-DE-1 and nZVI-DE-2 reached the decolorization of 90% and 98% of all dye. By contrast, controls such as nZVI and DE-1 and DE-2 removed 40%, 37%, and 24% of the dye. Toxicological analysis using zebrafish showed that AB causes a severe defect in development, and embryos die after exposure. However, the water samples treated with nZVI-DE-1 and nZVI-DE-2 are not harmful to the zebrafish embryos during the first 24 h. However, all embryos exposed to the new material for more than 48 hpf had cardiac edema, smaller eyes, and curved and smaller bodies with less pigmentation.

**Keywords:** nanoparticles; water treatment; dyes; zero-valent iron; acid blue

## 1. Introduction

Textile industries discharge into wastewater without any treatment around 100 tons of dyes worldwide [1]. The concentration of dyes in industrial effluents can reach 500 mg/L and pollute local freshwater, reducing the efficiency of sunlight and thereby impeding photosynthesis. [2]. As a result, the stream's water temperature decreases, and photoautotrophic organisms such as algae, euglena, and cyanobacteria can no longer survive [3]. The death of these organisms is an ecological loss because they play an essential role in the cycle of nutrients and can absorb organic matter present in the stream. They can also remove carbon dioxide from the atmosphere and are the base of the entire upper food chain [4]. Other parameter levels affected by the dye pollution are the biochemical oxygen demand (DBO), chemical oxygen demand (COD), and total suspended solids (TSS). These changes in the water quality composition could promote the death of fish and amphibians and change the microbial community [5]. Furthermore, the local population lost a valuable freshwater resource.

Acid blue (AB) is an anthraquinone dye, one of the most widely used colorants in the industrial sector, including cosmetics, food coloring, or dyeing different fibers. It is commonly mixed with sulfuric acid to make it more soluble before any industrial application, increasing toxicity [6]. AB is not only toxic for aquatic life but can also cause skin irritation, cornea damage, and promote the development of tumors and cancer [7]. Several kinds of research have been carried out with the aim of removing dyes from water using aerobic or anaerobic degradation, filtration, adsorption, membrane filtration, and other methods [8–10]. However, these processes have disadvantages, such as the extended time required for the treatment, high operational costs, and efficiency [11]. Dye pollution is not a novel problem; in fact, it is a persistent problem without a solution. The best solution made by some countries is to move their industries to poor lands lacking rules.

Recently, nanomaterials have been used to remove several pollutants, including dyes such as crystal violet [12], methyl orange [13], acid red 88, and Black 5 [14]. Nanoparticles have a relatively high surface-area-to-mass ratio, rendering them more reactive than conventional materials used for water treatment [15]. Iron is the material most used in several remediation treatments due to its low cost, abundance, ease, and reactivity. These materials remove dyes through an adsorption and reduction process [16]. However, this technology has to overcome some challenges to be applied successfully; during synthesis, the nanoparticles aggregate easily, thereby losing surface area [17,18] and becoming toxic for aquatic life [19]. To avoid these problems, some researchers added surfactants to nanoparticles or supported them [20]. The material used to support nanoparticles needs to be affordable, accessible, and compatible with the nanoparticles [21]. The diatomite earth is an affordable and well-known filter material in the industry, used to remove ammonium ions [22], heavy metals [23], and even some dyes such as Red 3BS and Yellow 5GF [24] using their absorption capacity. This material could be an exciting material to support the zero-valent iron nanoparticles and make an efficient filter scalable to an industrial application.

An optimal filter is affordable, removes high levels of the pollutant, and is safe to the user. Zebrafish (*Danio rerio*) has been established as an ideal model for toxicological studies to test the effects of contaminants such as alkaloids, glycosides, metals, alcohols, and carboxylic acids, among others [25–27]. Zebrafish have many characteristics that make them a good model for testing toxicity. Firstly, female zebrafish are able to produce hundreds of eggs, and embryos are transparent, which allows for a close observation of their development under a microscope. Secondly, the rapid growth of zebrafish compared to other vertebrates makes it an ideal model for high-throughput analysis [28].

Therefore, the objective of this research was to characterize and synthesize nanoscale zero-valent iron supported on diatomaceous earth and test if treated water had a negative effect on the viability of zebrafish embryos.

## 2. Materials and Methods

### 2.1. Preparation of Diatomite Earth and Synthesis of Nanomaterial

The diatomite earth (DE) was washed with 1 M HCl for 8 h under agitation (150 rpm). After the acid-washed, the DE was rinsed with water before using it as nZVI support. Two types of nZVI-DE were synthesized. The first identified as nZVI-DE-1 was prepared from 0.3 M $FeCl_2.4H_2O$ at a 40/60 proportion of $FeCl_2.4H_2O$/DE, while the second type identified as nZVI-DE-2 was made at a 50/50 ratio of $FeCl_2.4H_2O$/DE. nZVI and nZVI-DE-1 and -2 were obtained using the chemical reduction method of $FeCl_2.4H_2O$ in an aqueous solution using $NaBH_4$ as a reducing agent, due to its simplicity and efficiency in securing nZVI [29]. The $FeCl_2.4H_2O$ and DE were added to a previously ethanol deoxygenated by bubbling it with $N_2$ gas for 30 min. The iron salt was dissolved in 50 mL ethanol, and the solution was kept under $N_2$ bubbling and stirring for 30 min at 400 rpm at 25 °C. Later, 1.5 M $NaBH_4$ solution was slowly added into the $FeCl_2$/DE solution. After the reaction, the solution was kept stirring at 700 rpm for 60 min. In the end, both types of nZVI-DE were washed 10 times with ethanol and dried in an oven, bubbling with argon at 50 °C. The synthesis of the nZVI without DE support followed the same technique.

## 2.2. Characterization of Iron Nanoparticles Supported DE

The XRD analysis was carried out using a Bruker D8 Advance Eco diffractometer coupled with a copper source without a monochromator; the samples were placed in a sample holder with a 2θ range of 5–130° at a size and time of 0.02° and 0.2 s. The analysis of the crystalline phases was carried out using Match software.

A scanning electron microscope (SEM) HRSEM-AURIGA Zeiss integrated with an *X*-ray scattered energy (EDX) analyzer at a voltage of 1–10 keV at a high vacuum was used. Both the mapping and the elemental compositional analysis were carried out by selecting random areas using the EDX analyzer three times per sample. The samples were placed on graphite tape supported on metal discs.

A transmission electron microscope (TEM) JEM-ARM200F operated at a high vacuum was used to describe the morphology, size, and distribution of the particles. The samples were prepared by dispersing a small amount of powder in absolute ethanol with the aid of an ultrasonic bath. The dispersion (10 μL) was applied on a copper mesh for TEM of 300 mesh with Lacey/Carbon film and allowed to evaporate in a desiccator. We used Digital Micrograph to calculate the particle size.

The specific surface area was measured using the Brunauer–Emmett–Teller (BET) method of $N_2$ adsorption, using a Micromeritics Gemini 2360 surface area analyzer. Before analysis, the samples were degassed under vacuum at 70 °C for three hours.

## 2.3. Batch Degradation Experiments

The acid blue 52 is an anthraquinone dye with the following molecular formula $C_{22}H_{15}N_3NA_2O_9S_2$, a molecular weight of 575.48, and a CAS registry number of 61752-67-8. The experiment consisted of comparing two different relationships between the quantity of nZVI and DE (nZVI-DE-1 = 40/60, and nZVI-DE-2 = 50/50) but held the same amount of iron (62 mg) in every treatment.

The experiment was carried out using serum bottles with 50 mL of water contaminated with dye (1000 mg/L) loaded with 658 mg of Nano nZVI-DE. The controls were (i) DE-1 and DE-2 to evaluate dye degradation to adsorption process, (ii) nZVI to assess the effect of free nanoparticles without the support, and (iii) AD to calculate the abiotic degradation. The experiments were carried out at 25 °C with a mix of 150 rpm and pH 7.4. All experiments were performed in triplicate. The removal of AB was measured with the UV spectra (BioSpectrometer, Eppendorf) at 526 λ nm.

## 2.4. Fish Maintenance and Strains

AB line and wild-type zebrafish (*Danio rerio*) embryos were obtained from natural crosses and raised at 28 °C based on standard procedures [30]. Eggs were obtained by random pairwise mating of zebrafish. The following morning, the eggs were harvested and transferred into plastic Petri dishes (60 eggs per dish) containing 10 mL fresh embryo water. Further, unfertilized, unhealthy, and dead embryos were identified under a dissecting microscope and removed. Morphological criteria determined embryonic stages, according to Kimmel and collaborators [31]. Zebrafish were handled in compliance with local animal welfare regulations, and all protocols were approved by the ethics committee (Instituto de Biotecnología, UNAM, Mexico City, México).

## 2.5. Toxicological Studies

At 3.5 h postfertilization (hpf), embryos were again screened, and any additional dead and unhealthy embryos were removed. Embryos at the sphere stage were selected and transferred into a 48-well plate, with 10 in each well. Later, the embryo water was absorbed, and 300 μL of the treated water samples (AB dye, nZVI-DE-1, nZVI-DE-2, DE-1, DE-2) was loaded into each well; experiments were performed in triplicate. The control was a set of embryos grown with embryonic water in the same plate as well as in a petri dish independently. The embryonic plates were cultivated in a moist chamber at 28 °C from 6 to 24 hpf, and the viability of the embryos was observed under a dissecting microscope.

Embryos at 24 hpf were anesthetized with tricaine, immobilized with methylcellulose on agar plates and visualized with a stereomicroscope (Leica MZ 12.5), and photographed using a CCD camera (AxioCam MRc 5, Zeiss) and AxioVision Rel. 48 software. Statistical analysis was performed with three independent experimental replicates. The exact number of biological replicates is indicated in the figure legends. Statistical analysis was performed with Prism (GraphPad) Student's *t*-test. Error bars in column graphs represent the standard deviation of the mean (s.d.).

## 3. Results

### 3.1. Characterization of DE and Iron Nanoparticles

According to the results, DE presents a Brunauer–Emmett–Teller (BET) specific surface area of 23 $m^2/g$ with a pore volume of 0.56 $cm^3/g$. This result is in accordance with another research published previously (Crane and Sapsford, 2018). The homogenous pore volume allows the molding of narrow-sized particles dispersed homogeneously on their internal surfaces [32]. Regarding the composition, the SEM/EDX analysis of DE showed high levels of oxygen at around 52%, given that most of the materials that form DE are oxides, such as silicon oxide or aluminum oxide (Table 1).

**Table 1.** Element composition of the materials.

| Element | DE | nZVI | nZVI-DE-1 | nZVI-DE-2 |
|---|---|---|---|---|
| Si | 36.7 $\pm$ 1.1 | ND | 23.9 $\pm$ 2.9 | 23.3 $\pm$ 0.6 |
| Al | 8.6 $\pm$ 1.5 | ND | 8.4 $\pm$ 0.2 | 7.0 $\pm$ 0.3 |
| O | 52.1 $\pm$ 0.1 | 19.27 | 43.2 $\pm$ 2.2 | 42.1 $\pm$ 0.3 |
| S | 0.9 $\pm$ 0.1 | ND | 1.3 $\pm$ 0.5 | 1.2 $\pm$ 0.3 |
| K | 0.5 $\pm$ 0.1 | ND | 0.7 $\pm$ 0.3 | 0.7 $\pm$ 0.2 |
| Fe | ND | 80.73 | 21.9 $\pm$ 2.3 | 24.9 $\pm$ 0.5 |

The second most abundant element in DE is Si (37%), as was expected and elsewhere reported [33]. The Si content varies depending on the physicochemical conditions present during the formation of the DE bank [34]. These minor variations could affect the material's performance. The higher the content of Si, the more silanol groups available to react with polar organic compounds, increasing the number of compounds adsorbed by DE [35]. Carbonate minerals can also affect the number of compounds that DE can be absorbed, because the deposit of this material along the diatomite structure reduces its porosity [34].

The XRD analysis showed several crystal structures formed in the DE with the abundance of cristobalite, berlinite, calcite, Kaolinite, feldspar, and quartz (Figure 1a). Regarding nZVI-DE-1 and nZVI-DE-2, TEM images showed spherical forms in clusters between 20 and 40 nm, with an average size of 35 $\pm$ 8 nm (Figure 2a,b). The SEM/EDX analysis showed a decrease in the levels of Si and O compared with the DE analysis given that space is occupied by iron 22% and 25% for nZVI-DE-1 and nZVI-DE-2, respectively (Table 1). The XRD pattern of nZVI-DE showed diffraction peaks at the 2θ of 44.90°, confirming the presence of zerovalent iron in both treatments of nZVI-DE and on nZVI treatment (Figure 1b). Crane and Sapsford et al. [36] have reported similar XRD patterns of iron and DE. However, they indicate the presence of other metals such as Al, suggesting that perhaps the duration of the acid wash ~2 h was not enough.

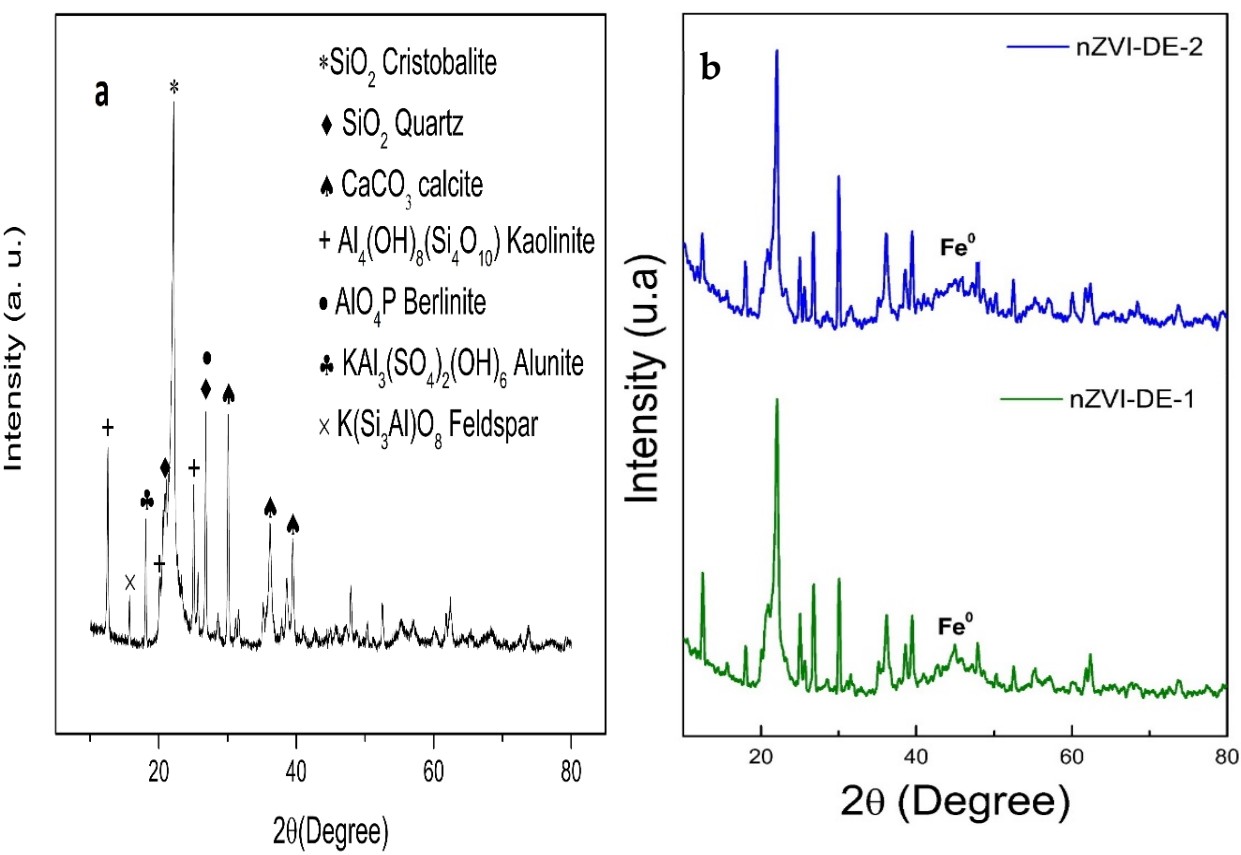

**Figure 1.** (**a**) XRD pattern of diatomaceous earth. (**b**) XRD pattern of nZVI-DE-1 and nZVI-DE-2 treatments.

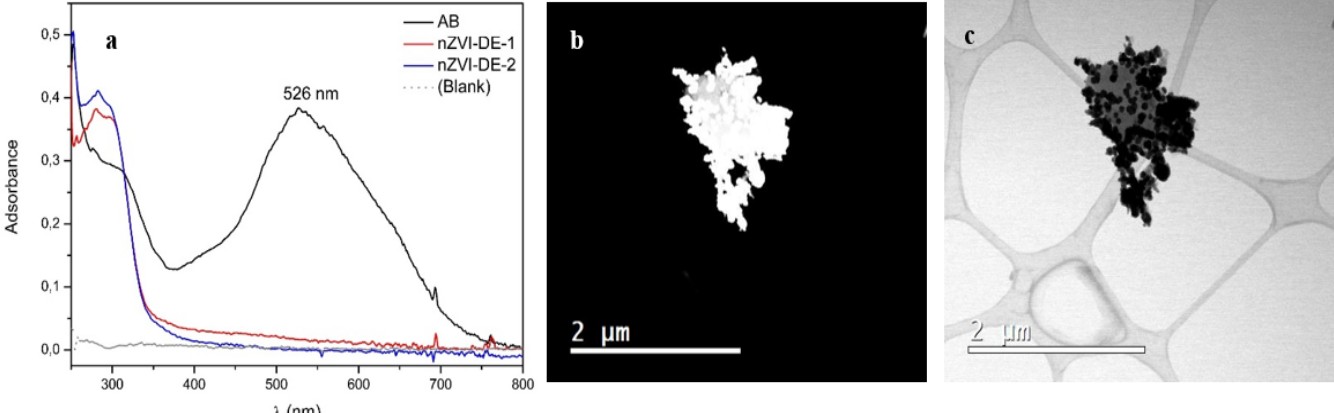

**Figure 2.** (**a**) Absorbance signal of the dye and the dye removed by the nZVI-DE-1 and -2, while dark field (**b**) and bright field (**c**) micrographs of the Fe nanoparticles decorating DE particles. Fe nanoparticles brighter on dark field contrast with those of the bright field.

### 3.2. Batch Degradation Experiments

The reaction proceeded in the first 5 min with a drastic change of color, and later, the difference decelerated and stopped 7 min later. Figure 2a shows the UV-vis spectra taken 10 min after, when the reaction was stable. The black line shows the spectra of the water with 1 g/L of AB, the red line indicates the nZVI-DE-1 treatment, which removed 90% of the pigment, and the blue line shows that the nZVI-DE-2 treatment achieved 98% removal of AB. The nZVI control removed just 40% of AB, and DE-1 and DE-2 controls removed 37% and 24%, respectively.

The contrast between the catalytic power of supported and unsupported nanomaterials is significant: nZVI-DE-1 and nZVI-DE-2 removed 50 and 58% more dye than

unsupported nanoparticles. In nZVI-DE-1 and nZVI-DE-2, the nZVI particles removed the dye through reduction and adsorption processes [13,37]. At the same time, the DE played a protective role for the nZVI catalyst (reducing agglomeration) and contributed to the adsorption of AB. The amount of AB removed and adsorbed could be calculated based on removal results of nZVI and DE controls, and evidently, around 40–50% of the removal was adsorbed by the DE, and the rest was the reduction performed by the nanoparticles. Another factor that could contrast the removal of both treatments could be the amount of DE. nZVI-DE-1 had less DE where nanoparticles were supported, and this treatment also had less iron than nZVI-DE-2.

The removal performed by the nanoparticles in contact with the contaminant occurs because the nanoparticles play a role as electron mediators; moreover, H atoms are generated, which causes the break of the double bond (-C=C-), eliminating the chromophore group of the dye [38]. By destroying one of the critical components of the dye, the absorption peak at a wavelength of 526 nm was reduced as a result. At the same time, $Fe^{\circ}$ reacted and formed oxides such as $FeO_2$, $FeO_3$, and $FeOH$, which have a high capacity to absorb some molecules of the pollutant [39]. However, this capacity to absorb could be affected by internal changes in pH [40]. The pH of the dye solution plays an essential role in the entire adsorption process and the adsorption capacity, influencing loads of the nZVI-DE and DE. This changes the degree of ionization and dissociation of the functional groups in the active sites of the adsorbent materials [41]. In the beginning, the pH of the treatments was 7.4. However, at the end of the removal experiments, nZVI-DE-1 had 8.9, and nZVI-DE-2 had 8.7, while nZVI had a more neutral pH of 7.7 and DE-1 and -2 had an acid pH of 4 and 4.6, respectively. The treatment with nanoparticles showed pH values around 8 because the dye removal is a dynamic process where $Fe^{\circ}$ nanoparticles begin to be transformed into oxides as $Fe^{2+}$, $Fe^{3+}$, $Fe(OH)_3$, and $Fe(OH)_2$, which reduces the quantity of $H^+$ and raises the pH of the liquid [42], whereas in the treatment with only DE, the pH of the solutions finishes acidic due to the protonation of surface silanol groups where protons are forming conjugate acids that lower the pH [43,44].

### 3.3. Toxicological Analysis of Treated Water on Zebrafish Embryos

Embryos were treated with AB, and the water was treated with nZVI-DE-1, nZVI-DE-2, DE-1, and DE-2 during 24, 48, 72, and 96 h postfertilization (hpf). Embryos exposed only to AB died after treatment; meanwhile, embryos treated with nZVI-DE-1 and nVZI-DE-2 developed alongside the embryos in control water (Figure 3A–C,G). Embryos grown in the presence of DE developed regularly (data not shown). The embryos treated with nZVI-DE-1 and nZVI-DE-2 were allowed to develop until 24 hpf; embryos developed normally, but a slight developmental delay was observed compared to the control (Figure 3D–F). However, after 24 h of culture, particles associated with the chorion were visible in nZVI-DE-1 and nZVI-DE-2 samples, possibly due to an agglomeration of nanoparticles (Figure 3B,C asterisk), and perhaps, this caused the developmental delay observed in these samples. At 48 and 72 hpf (data not shown), there was no difference in the viability of the embryos when compared to the control (Figure 3N). Still, an unambiguous effect in the morphology of the embryos was observed as all of the embryos had cardiac edema, smaller eyes, and curved and smaller bodies with less pigmentation (Figure 3H–O). We also observed particles attached to the chorion (Figure 3A–C,J); these could be the nanoparticles since we did not observe these aggregates in the control embryos. At 96 hpf, the embryos treated with the water from nZVI-DE-1 and nZVI-DE-2 died; meanwhile, control embryos developed normally. The delay in development at 24 hpf; the morphological effect at 48 and 72 hpf and the comprised viability of embryos at 96 hpf could be due to the presence of the nanoparticles. Previous work conducted by Almeida et al. [45] showed that embryos exposed to iron micro- and nano-particles had effects.

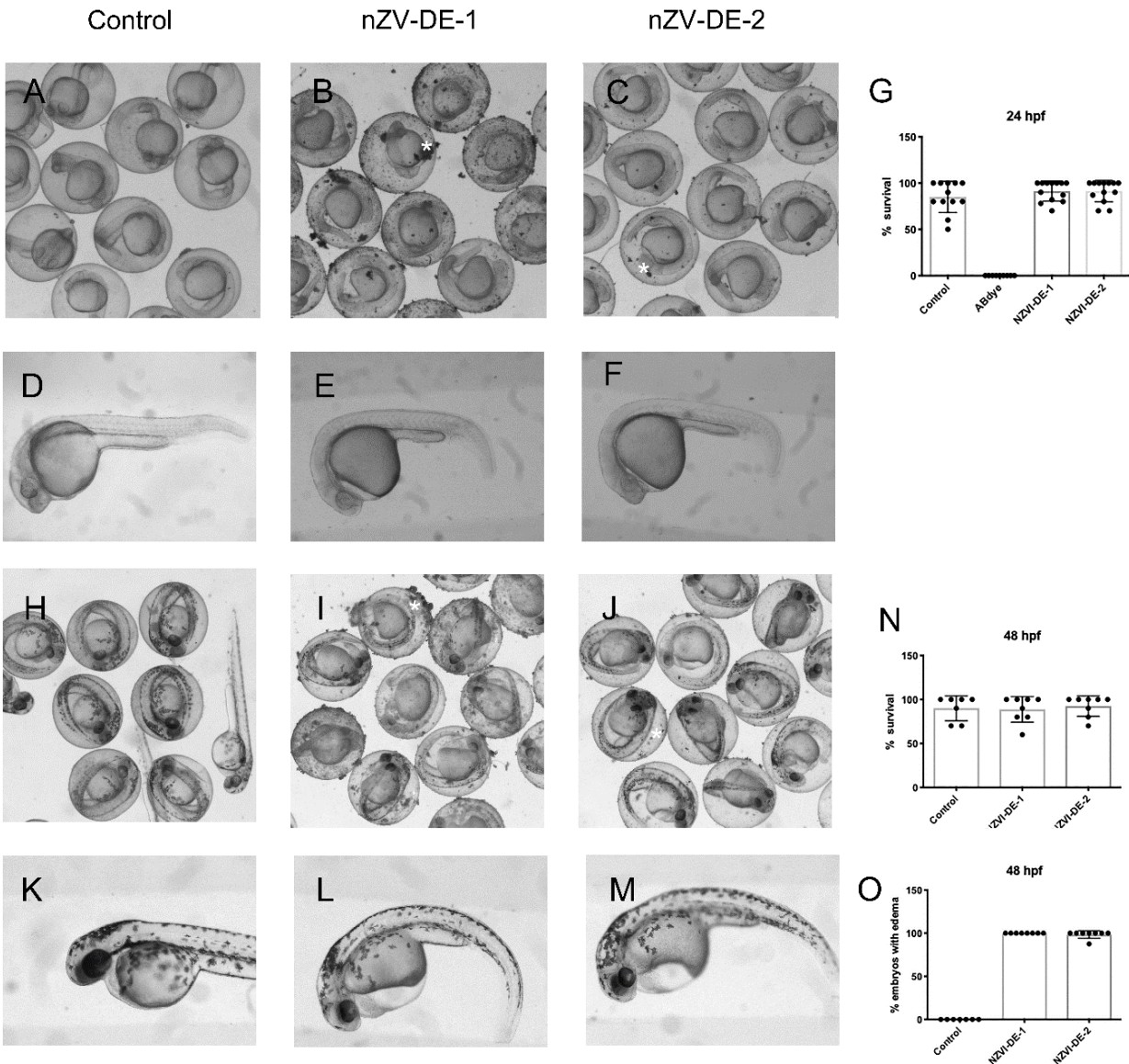

**Figure 3.** Zebrafish embryonic development progress in the presence of nZV-DE-1 and nZV-DE2. (**A**). Control embryos at 24 hpf. (**B**,**C**). Embryos at 24 hpf proceed with development when exposed to contaminated water is treated with nZV-DE. Agglomeration of nanoparticles can be observed in the chorion (asterisk in (**B**,**C**)). (**D**–**F**). A slight delay in development is observed in nZV-DE-1 (**E**) and nZV-DE-2 (**F**) embryos compared to the control (**D**). (**G**,**N**) Viability is fully restored in nZV-DE treated water exposure compared to AB dye exposure. (**H**) Control embryos at 48 hpf. (**I**,**J**) Embryos developed until 48 hpf as the control embryos; nevertheless, the embryos treated with nZV-DE (**L**,**M**) present cardiac edema (**O**), smaller eyes curved body plan, less pigmentation, and are smaller compared to the control (**K**).

Nevertheless, the improvement of embryonic viability was highly significant compared to the embryos treated only with the AB dye (Figure 4). In order to determine the exact moment when the AB had an effect, dye cultures of 4 h were carried out. Here, we could observe that after 4 h of treatment, all embryos treated with AB dye were already dead; meanwhile, control embryos and those treated with nZV1-DE-1 and nZV1-DE-2 developed normally (Figure 4). Additionally, the embryos treated with AB dye were colored as was the chorion (Figure 4D,H), while nZV1-DE-1 and nZV1-DE-2 treated embryos followed normal development without coloration (Figure 4). Our results suggest that the treatment of nZV1-DE-1 or nZV1-DE-2 is sufficient to avoid the lethality observed in embryos exposed to AB dye, given that development generally proceeds at the first hours of treatment. Nevertheless, further stages of development, such as 48 and 72 hpf, were

affected as a result of the nanoparticles present in the samples. Potential applications of nZVI-DE could mitigate the effects observed in the embryos after 24 h via filters where the treated water is in contact with the organism for a short period.

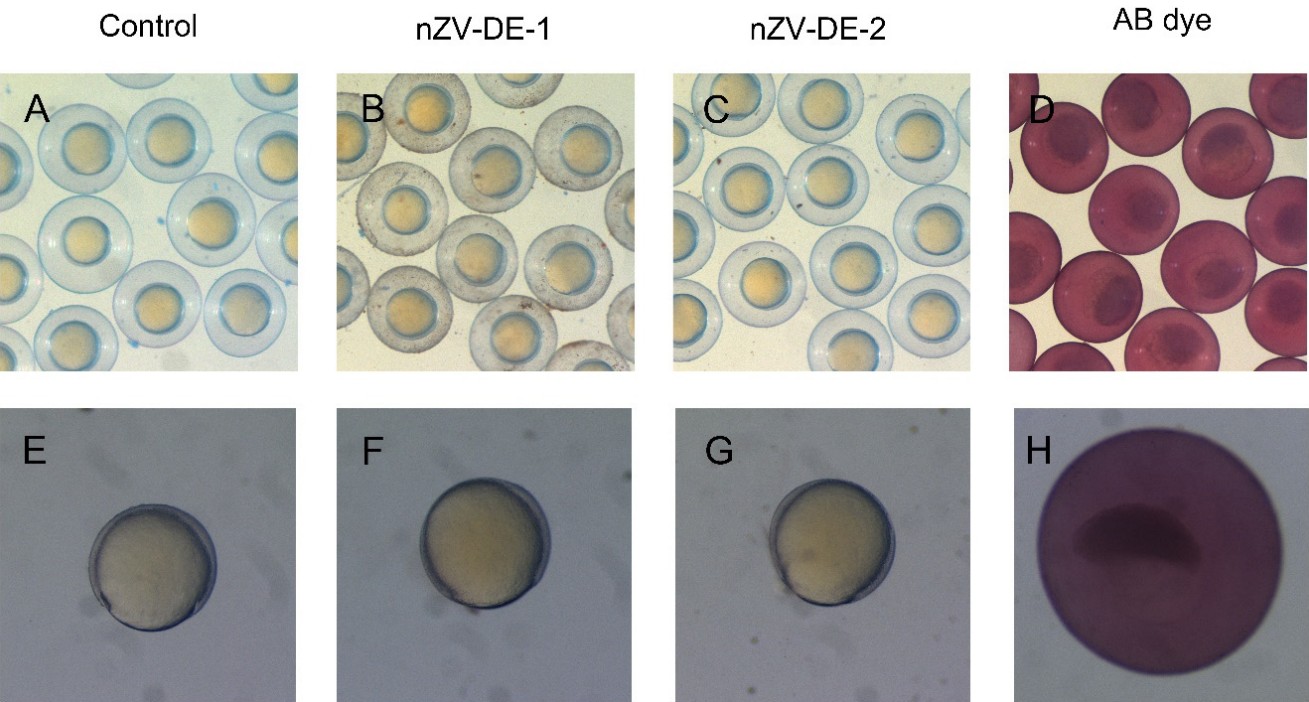

**Figure 4.** Early zebrafish development is restored after nZV-DE treatment. (**A**). Control embryos after 4 h of treatment. (**B**) Embryos treated with nZV-DE-1 develop normally. (**C**). Embryos treated with nZV-DE-2 develop as the control embryos. (**D**) Embryos treated with AB dye died after 4 h of treatment. (**E–H**) Higher magnifications show that embryos treated with nZV-DE (**F,G**) develop as the control embryos (**E**); meanwhile, embryos treated with AB dye absorb the dye and do not develop further and die.

## 4. Discussion

There is an increasing interest in using nanoparticles to remove different kinds of pollutants and implementing this technology as a standard filter material; however, the possible risk to aquatic life or food chains is not sufficiently clear. The design of new materials to filters needs to be accompanied with toxicological assessments.

The nZV-DE has good performance because the material was highly efficient, removing the AB dying in a short time. Typically, biological treatments take at least 30 days to reach removals of around 60%. Furthermore, this material is superior to conventional DE filtration in addition to the removal efficiency; the treated water also does not have an acid pH. However, the toxicological results showed all embryos exposed to the new material for more than 48 hpf had cardiac edema, smaller eyes, and curved and smaller bodies with less pigmentation. One possible solution to implement nZV-DE material to filters is combine the treatment with a second filter that retain the nanoparticles debris. Although this material is very promising, more research is needed to deeply elucidate the toxicological effects, emulating different kinds of contacts, because if this treated water is to be released into the river, the aquatic life, such as fish, will be exposed to this water for a short period of time.

## 5. Conclusions

The nZVI-DE-1 and nZVI-DE-2 successfully removed more than 90% of one of the most widely used pollutants, acid blue, in a short time. The toxicology test of the treated water showed that wild-type zebrafish (*Danio rerio*) embryos developed entirely normally during the first 24 h. However, after 48 h, all of the embryos had cardiac edema, smaller

eyes, and smaller bodies with less pigmentation than the control sample. Consequently, we suggest that nanoparticles technologies use supported materials and that the nanoparticle technology used to remove pollutants has limited contact with streams to avoid adverse effects on aquatic life.

**Author Contributions:** Conceptualization, L.B.-D.; methodology, L.B.-D, D.S., E.F.-R., H.M.P.-V., and O.S.-F.; software, L.B.-D, E.F.-R., E.J.-C. and D.S.; validation, E.F.-R. and D.S.; formal analysis, E.J.-C., D.S. and E.F.-R.; resources, H.M.P.-V. and O.S.-F.; data curation, L.B.-D.; writing—original draft preparation, L.B.-D.; writing—review and editing, L.B.-D., D.S., E.F.-R., H.M.P.-V. and O.S.-F.; supervision, L.B.-D. All authors have read and agreed to the published version of the manuscript.

**Funding:** This research received no external funding.

**Institutional Review Board Statement:** The study was conducted in compliance with local animal welfare regulations and approved by Institute's Ethical Committee (Instituto de Biotecnología, UNAM) protocol 412-30 September 2021.

**Informed Consent Statement:** Not applicable.

**Acknowledgments:** The authors thank ZEOLITECH for donating diatomaceous earth, Daniel Bahena Uribe, and Jorge Roque De La Puente of LANE, CINVESTAV, for their excellent technical help and advice on the characterization of nanoparticles, and Erick Justo Cabrera for their support in the nanoparticles synthesis. LB-D and DS thank the CICA program and Instituto de Biotecnologia UNAM for their support. In addition, LB-D thanks the Consejo Nacional de Ciencia y Tecnología (CONACYT) and their program CATEDRAS for supporting Project 285.

**Conflicts of Interest:** The authors declare no conflict of interest.

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
