# Peer review of "Using Nano Zero-Valent Iron Supported on Diatomite to Remove Acid Blue Dye: Synthesis, Characterization, and Toxicology Test"

_sustainability, doi:10.3390/su132413899_

Round 1

Reviewer 1 Report

For textile industry, acid blue is not a novel dye.

The DE-nZVI should be clarified its novelty. And the role and contribution of DE and n-ZVI during the removal of acid blue should be clarified.

The adsorption of acid blue on DE and n-ZVI, the reduction of acid blue by n-ZVI should also be clarified .

The experiments of the removal of acid blue was too simple, lots of factors should be carried out.

Author Response

Dear Reviewer, thanks for all your comments. We appreciate the time and effort that you dedicated to providing feedback on our paper.

Review 1

For textile industry, acid blue is not a novel dye.

Author response: We agree with the reviewer’s assessment. We have added some words and lines in order to clarify this point.

The revised text reads as follow on:

L51 Acid blue (AB) is an anthraquinone dye one of the most widely used colorants in the industrial sector

L59  Dye pollution is not a novel problem is a persistent problem without a solution. Until now, the best solution made for some countries is to move their industries to poor lands lacking rules.

The DE-nZVI should be clarified its novelty. And the role and contribution of DE and n-ZVI during the removal of acid blue should be clarified.

Author response: please reviewer the follow lines

L65-68

Iron is the material most used in several remediation treatments due to its low cost, abundance, ease, and reactivity. The iron nanoparticles removal dyes through an adsorption and reduction process [16]

L240-247

The contrast between the catalytic power of supported and unsupported nanomaterials is significant, nZVI-DE-1 and nZVI-DE-2 removed 50 and 58 % more dye than unsupported nanoparticles. In nZVI-DE-1 and nZVI-DE-2, the nZVI particles were responsible for the removal of the dye through reduction and adsorption processes [13,37], while the DE played a protective role for the nZVI catalyst (reducing agglomeration) and contributed to the adsorption of AB. The amount of AB removed and adsorbed could be calculated based on removal results of nZVI and DE controls, and evidently, around 40 – 50 % of the removal was adsorbed by the DE and the rest was the reduction performed by the nanoparticles.

L250-263

The removal performed by the nanoparticles in contact with the contaminant occurs because the nanoparticles play a role as electron mediator; also, H atoms are generated and this causes the break of the double bond (-C = C-) eliminating the chromophore group of the dye [38]. Destroying one of the critical components of the dye and, as a result, the absorption peak at a wavelength of 526 nm was reduced (Figure 4a). At the same time, Fe° reacted and formed oxides such as FeO2, FeO3, and FeOH which have a high capacity to absorb some molecules of the pollutant [39]. However, this capacity to absorb could be affected by internal changes in pH [40]. The pH of the dye solution plays an essential role in the entire adsorption process and the adsorption capacity, influencing the loads of the nZVI-DE and DE. This changes the degree of ionization and dissociation of the functional groups in the active sites of the adsorbent materials [41]. At the beginning, the pH of the treatments was 7.4, however at the end of the removal experiments, nZVI-DE-1 had 8.9; and nZVI-DE-2 had 8.7; while nZVI had a more neutral pH of 7.7; and DE-1 and 2 had an acid pH of 4 and 4.6., respectively

L264-268

The treatment with nanoparticles showed pH values around 8 because the dye removal is a dynamic process where Fe° nanoparticles begin to be transformed into oxides as Fe2+, Fe3+, Fe(OH)3 and Fe(OH)2, which reduces the quantity of H+ and raises the pH of the liquid [42]. Whereas in the treatment with only DE, the pH of the solutions finishes acidic due to the protonation of surface silanol groups where protons are forming conjugate acids that lower the pH  [43,44].

The adsorption of acid blue on DE and n-ZVI, the reduction of acid blue by n-ZVI should also be clarified.

Author response: we don’t measure the adsorption of acid blue.

The experiments of the removal of acid blue was too simple, lots of factors should be carried out.

Author response: I don’t understand the question or comment or advice. The Material and Methods section specify all the factors that we consider

Reviewer 2 Report

The authors treated diatomaceous earth (DE) with 1M HCl and then prepared a composite of DE with zero-valent iron (ZV) by reducing Fe2+ in ethanol with NaBH4 reduction at mass fraction ratios of 0.4/0.6 (Fe2+/DE) and 40/60 (Fe2+/DE). The obtained NZV-DE composites were characterized by BET, XRD, SEM/EDX, TEM and were investigated for acid blue removal. The toxicity of the NZV-DE towards Danio rerio was investigated, as well as the toxicity of acid blue treated with NZV-DE.

The manuscript presented is in need of significant improvement, there are some shortcomings which will be addressed shortly. My main objection, however, is that the manuscript lacks sufficient novelty. The toxicity of NZV is well known (A. Keller et al., Toxicity of Nano-Zero Valent Iron to Freshwater and Marine Organisms, PloS One 7 (2012) e43983). A plethora of publications dealing with dye degradation by NZV is available. Even more results on the toxicity of NZV towards Danio rerio can be found in publicly available thesis and dissertations, namely H.G. Bulovsky, The stability, toxicity and reactivity of zero valent iron nanoparticles and A.H. Schiwy, The nanotoxicology of a newly developed zero-valent iron nanomaterial for groundwater remediation and its remediation efficiency assessment combined with in vitro bioassays for detection of dioxin-like environmental pollutants. From the results presented in the manuscript, it is not possible to determine whether the toxicity towards Danio rerio is exhibited by dislodged/free NZV particles in solution, or whether due to the presence of NZV-DE itself only. I believe a more thorough characterization of the NZV-DE suspension should be a good idea, namely with DLS.

My specific comments related to the manuscript are as follows:

  • The statement that simple prokaryotes and eukaryotes are an indispensable food source for “several organisms”, page 1 lines 41-34 is a serious understatement, as the entire upper food chain relies on these organisms as a foundation.
  • DQO, page 1 line 44, is an unusual abbreviation of chemical oxygen demand. I suspect DQO comes from Spanish language, please use COD instead.
  • Please specify the exact Colour Index International classification for the acid blue dye used, as “acid blue” encompasses an entire spectrum of acidic anionic dyes and is not specific The dye used may belong to the anthraquinone or triarylmethane class of acid blue. The recorded toxicity response is strongly dependent on the exact pollutant investigated.
  • The experimental design for batch degradation experiments should be elaborated further. Factorial analysis should indicate that a statistical experimental design was utilized, which seems not to be the case. Please amend this.
  • At what pH were the batch degradation experiments carried out?
  • Please provide BET isotherms in the supplementary materials.
  • Please provide a meaningful description for Table 1.
  • Elemental analysis obtained by EDX should be amended with EDX mapping, if possible/allowed by EDX analysis software. If only single point analyses were performed, how was the relative error for each element calculated?
  • Why were no SEM micrographs, along with TEM micrographs?
  • No results related to COD were provided or discussed, despite COD measurements being mentioned in the experimental section.

Author Response

Dear Reviewer, thanks for all your comments. We appreciate the time and effort that you dedicated to providing feedback on our paper.

  • The statement that simple prokaryotes and eukaryotes are an indispensable food source for “several organisms”, page 1 lines 41-34 is a serious understatement, as the entire upper food chain relies on these organisms as a foundation.

Author response: Thank you for pointing this out. The reviewer is correct, and we have made a change in that line.

The revised text reads as follows on L44, they can also remove carbon dioxide from the atmosphere and are the base of the entire upper food chain.

  • DQO, page 1 line 44, is an unusual abbreviation of chemical oxygen demand. I suspect DQO comes from Spanish language, please use COD instead.

Author response: Thank you for pointing this out. We have made the change to COD in L47.

  • Please specify the exact Colour Index International classification for the acid blue dye used, as “acid blue” encompasses an entire spectrum of acidic anionic dyes and is not specific The dye used may belong to the anthraquinone or triarylmethane class of acid blue. The recorded toxicity response is strongly dependent on the exact pollutant investigated.

Author response: As suggested by the reviewer, we have added more information about the dye

L131 The acid blue 52 is and Anthraquinone dye with the following molecular formula C22H15N3NA2O9S2, a molecular weight of 575.48 and a CAS registry number of 61752-67-8.

  • The experimental design for batch degradation experiments should be elaborated further. Factorial analysis should indicate that a statistical experimental design was utilized, which seems not to be the case. Please amend this.

Author response: Thank you for pointing this out. The section has been corrected

The revised text reads as follows on L137-140: The experiment consisted of compare two different relationships between the quantity of  nZVI and DE (nZVI-DE-1 = 40/60, and nZVI-DE-2 = 50/50) but held the same amount of iron (62 mg) in every treatment.

  • At what pH were the batch degradation experiments carried out?

Author response: the information about the pH of the treatment is in L252 and we have added that information in material and methods L143

L143 The experiments were carried out at 25° C with a mix of 150 rpm and pH 7.4.

L252 At the beginning, the pH of the treatments was 7.4, however at the end of the removal experiments, nZVI-DE-1 had 8.9; and nZVI-DE-2 had 8.7; while nZVI had a more neutral pH of 7.7; and DE-1 and 2 had an acid pH of 4 and 4.6., respectively.  The treatment with nanoparticles showed pH values around 8 because the dye removal is a dynamic process where Fe° nanoparticles begin to be transformed into oxides as Fe2+, Fe3+, Fe(OH)3 and Fe(OH)2, which reduces the quantity of H+ and raises the pH of the liquid [41]. Whereas in the treatment with only DE, the pH of the solutions finishes acidic due to the protonation of surface silanol groups where protons are forming conjugate acids that lower the pH  [42,43].

  • Please provide a meaningful description for Table 1.

Author response: Thank you for pointing this out. We have made the change in L181

The revised line reads as follow: Table 1. Element composition of the materials.

  • Elemental analysis obtained by EDX should be amended with EDX mapping, if possible/allowed by EDX analysis software. If only single point analyses were performed, how was the relative error for each element calculated?

Author response: We made the EDX mapping using an EDX analysis software. At least 3 measurements to give an averaged elemental composition were made, also standard deviations were calculated and indicated in Table 1.

L11-115:

A scanning electron microscope (SEM) HRSEM-AURIGA Zeiss integrated with an X-ray scattered energy (EDX) analyzer at a voltage of 1-10 keV at a high vacuum was used. Both the mapping and the elemental compositional analysis were carried out by selecting random areas using the EDX analyzer three times per sample. The samples were placed on graphite tape supported on metal discs.

  • Why were no SEM micrographs, along with TEM micrographs?

Author response: Our TEM equipment have a better resolution than aur SEM equipment.

  • No results related to COD were provided or discussed, despite COD measurements being mentioned in the experimental section.

Author response: We don’t measure COD during the experiments. COD was mentioned in the introduction section when we showed the physicochemical changes the made the dye pollution in the streams.

Reviewer 3 Report

See attachment

Author Response

Dear Reviewer, thanks for all your comments. We appreciate the time and effort that you dedicated to providing feedback on our paper.

  1. Page 1, line 43: “parameters levels”, please check again. I think that it should have no S with parameters.

Answer: Page 1, line 45, now reads: “parameter levels”

  1. Citation style in text is not unified. Sometimes, they use numbering and some time, they use name and year of publication. Please see Page 2, line 73; page 3, line 138; page 4, line 172, 174; and a whole manuscript.

DONE

Page 2, line 75 now reads: carboxylic acids, among others [26–28]

Page 3, line 142 now reads: Kimmel and collaborators [30]

Page 4, line 176 now reads: during the formation of the DE bank [33].

Page 4, line 179 now reads: number of compounds adsorbed by DE [34]

Page 7, line 272 now reads: Previous work done by Almeida et al. [44] showed…

  1. Writing style of names of samples are also different from one to one. For example, sometimes, they use “nZVI-DE-2” and sometimes, they use “NZVI-DE-2”. Please unify the style. Page 2, line 88; label of Figure 1, 2; and so on.

DONE

Page 3, line 91 now reads: identified as NZVI-DE-2 was made at a 50/50 …

Label of Figure 1,2 now reads: Figure 1. (a) XRD pattern of diatomaceous earth. (b) XRD pattern of NZVI-DE-1 and NZVI-DE-2 treatments

  1. Sometimes, they use time unit as “min.” and sometimes, they use “min”. Please unify it. See Page 5, line 194.

DONE

Page 5, lines 198-200 now reads: The reaction proceeded in the first 5 minutes with a drastic change of color, and later, the difference decelerated and stopped 7 minutes later. Figure 2a shows the UV-vis spectra taken 10 minutes after

  1. Please double check the unit of degree of 2 theta (o). Page 3, line 102

DONE

Page3, line 106 now reads:

  1. Materials and Methods:

2.2 Characterization of ion nanoparticles. I think it should be named “Characterization of iron nanoparticle supported DE”.

DONE

In this section, there are many short subtitles (e.g. 2.2.1~2.24.). I think that combining them together to one subtitle is better. Shorten the detail process of characterizations because they are common information.

We have renamed the 2.2 section as you suggest “Characterization of iron nanoparticle supported DE” and eliminated all the short subtitles.

  1. Page 4, line 145: ul should be µl

DONE

Page 4, line 150 now reads: absorbed, and 300 µl of the treated water samples

  1. Sometimes, they use “Fig.” and sometimes, they use “Figure”. Please unify it.

DONE

  1. Page 6, line 146: H+ or H+

DONE,

Page 6, line 246: H+

  1. b) Scientific comments:
  2. Add story about reason for selecting zero valence iron nanoparticles in this study at the introduction part.

Author response: As suggested by the reviewer, we have added more information about iron nanoparticles

Page 2, lines 66- 69 Iron is the material most used in several remediation treatments due to its low cost, abundance, ease, and reactivity. The iron nanoparticles removal dyes through an adsorption and reduction process.

  1. XRD analysis is not clear. Some peaks were not indexed. In addition, one phase has only one peak indexed. Commonly, it should have several peaks belong to one phase. Please index it more precisely.

DONE (FIG. 1a)

  1. Authors mentioned “However, the XRD analysis showed several crystal structures formed in the DE with the abundance of cristobalite, tridymite, and quartz, but anyone with carbonate, which ensures the condition of the material to be a practical support to the nanoparticles (Fig. 1a).”. Does it mean that many phases are better for this application? Please make it clear and give reference.

We have re-indexed the XRD pattern of the diatomaceous earth sample. The three main peaks of calcite have been identified. We have eliminated the text regarding “condition of materials to be a practical support”

  1. Authors should include the UV-abs spectra of reaction with nanoparticles and only DE in Figure 2a because those already mentioned in text.

We eliminate “Figure 2” at the end of the line 239

  1. 2a should added reaction time to figure’s caption.

Dear reviewer, we don’t have these files. My computer was stolen, and the other team member, Erick, is working without his laptop in USA. We can repeat the UV-abs of a new sample.

Round 2

Reviewer 1 Report

The removal of dye by n-ZVI include adsorption and reduction. So, the author should do experiments to distinguish between adsorption and reduction.

The experiments of the removal of acid blue was too simple, lots of factors should be carried out.  The question means, The effect of different factors (the initial pH, the initial concentration of dye, the initial dosage of n-ZVI, the temperature, the initial oxygen) should and could be carried out.

Author Response

Dear review,

I agree with you. The experiments could have a lot of factors. However, we can not make more experiments at this moment. It's Ok if you feel this draft can not be published.  

Reviewer 2 Report

The revised manuscript has duly corrected previous inadequacies. I must point out that the written English can be improved. I do not, however, consider that the present state of the manuscript significantly affects the overall quality of the work.

Author Response

Dear review, 

Thanks for the comments regard to english. We have checked.

Reviewer 3 Report

Manuscript was significantly revised.

Author Response

Dear Reviewer 3

Thanks for the report.